# De novo adipocyte differentiation from Pdgfrβ+ preadipocytes protects against pathologic visceral adipose expansion in obesity

Mengle Shao [1], Lavanya Vishvanath[1], Napoleon C. Busbuso[1], Chelsea Hepler[1], Bo Shan[1], Ankit X. Sharma[1], Shiuhwei Chen[1], Xinxin Yu[1], Yu A. An [1], Yi Zhu[1], William L. Holland [1] & Rana K. Gupta[1]

Pathologic expansion of white adipose tissue (WAT) in obesity is characterized by adipocyte hypertrophy, inflammation, and fibrosis; however, factors triggering this maladaptive remodeling are largely unknown. Here, we test the hypothesis that the potential to recruit new adipocytes from Pdgfrβ+ preadipocytes determines visceral WAT health in obesity. We manipulate levels of *Pparg*, the master regulator of adipogenesis, in Pdgfrβ+ precursors of adult mice. Increasing the adipogenic capacity of Pdgfrβ+ precursors through *Pparg* over-expression results in healthy visceral WAT expansion in obesity and adiponectin-dependent improvements in glucose homeostasis. Loss of mural cell *Pparg* triggers pathologic visceral WAT expansion upon high-fat diet feeding. Moreover, the ability of the TZD class of antidiabetic drugs to promote healthy visceral WAT remodeling is dependent on mural cell *Pparg*. These data highlight the protective effects of de novo visceral adipocyte differentiation in these settings, and suggest Pdgfrβ+ adipocyte precursors as targets for therapeutic intervention in diabetes.

[1] Touchstone Diabetes Center, Department of Internal Medicine, University of Texas Southwestern Medical Center, Dallas, TX 75390, USA. Correspondence and requests for materials should be addressed to R.K.G. (email: Rana.Gupta@UTSouthwestern.edu)

Obesity confers significant risk for developing insulin resistance and type 2 diabetes; however, some obese individuals are relatively resistant to developing metabolic disease, at least for a period of time. This has long suggested that factors beyond elevated adiposity per se drive impairments in nutrient homeostasis. One clear determinant of metabolic health in obesity is the anatomical location of accumulated WAT. The accumulation of WAT in the intra-abdominal region (visceral adiposity) confers significant risk for developing metabolic disease in obesity whereas preferential expansion of WAT in subcutaneous regions (subcutaneous WAT) appears protective[1–3]. It was originally believed that the location of visceral WAT itself likely mediates some of its detrimental effects on energy metabolism[4,5]. Transplantation studies, cellular studies, and gene expression analyses suggest that factors intrinsic to these adipocytes may also contribute[6–12]. However, studies from Klein and colleagues suggest that factors beyond visceral WAT mass per se contribute to the development of insulin resistance[13,14]. As such, it has remained unclear how visceral WAT expansion leads to insulin resistance.

The mechanism by which individual WAT depots expand has also been proposed to be a critical determinant of insulin sensitivity in the setting of obesity[15–19]. In principle, the expansion of fat mass can occur by adipocyte hypertrophy (increased cell size) or adipocyte hyperplasia (increased adipocyte number through de novo differentiation from progenitors)[20]. Adipocyte hypertrophy is a defining feature of pathologic obesity[18,21–26]. The prevailing hypothesis is that as "overworked" fat cells reach their storage capacity, adipocyte death, inflammation, and fibrosis ensues. This is often associated with the deleterious accumulation of lipids in the liver, skeletal muscle, pancreas, and heart (termed "lipotoxicity"). Importantly, serum levels of the adipose-derived insulin-sensitizing hormone, adiponectin, drop considerably in pathologic obesity[17]. This hormone remains one of the best predictors of insulin sensitivity.

A "healthy" WAT phenotype is often observed in the subset of obese individuals who maintain insulin sensitivity or in those treated with the thiazolidinedione (TZD) class of anti-diabetic drugs. Healthy WAT is characterized by the presence of smaller and more numerous adipocytes, suggestive of tissue expansion by increased cell number. This phenotype is often associated with lower levels of tissue inflammation and fibrosis, along with a preservation of serum adiponectin levels. Based largely on these clinical observations, it has long been hypothesized that increasing the number of adipocytes, at the expense of adipocyte hypertrophy, could prevent pathological remodeling and adipose dysfunction[18,22,25,27–29]. Indeed, primary adipose stem cells isolated from obese individuals with healthy WAT exhibit a lower propensity to undergo adipocyte differentiation in vitro when compared to cells isolated from pathological WAT[30]. Whether this differentiation defect is a primary driver of pathological WAT expansion has been unclear.

Numerous animal models exhibiting increased adipocyte hyperplasia and reduced adipocyte hypertrophy have been described; data from these models motivated and/or significantly strengthened this overall hypothesis. This includes adipose-specific transgenic animals expressing Glut4, adiponectin, mitoNEET, and tenomodulin[31–34]. All of these models are more or equally obese than their control littermates, but metabolically healthy; there is considerable adipocyte hyperplasia, low inflammation, very little ectopic lipid accumulation, and relatively better insulin sensitivity than control animals. However, in all of these particular models, the Fabp4 or adiponectin promoter is used to drive constitutive transgenic expression from the onset of fetal WAT development. Importantly, these transgenes are active in differentiating/mature adipocytes. Therefore, whether the healthy

local and systemic phenotypes are due to increased adipocyte number per se, or rather due to improved function of engineered mature adipocytes remains unclear. Moreover, the constitutive activity of the transgenes limits the ability to address the importance of de novo adipocyte differentiation specifically in adult mice.

We recently reported that visceral adipocytes emerging in association with HFD feeding originate, at least in part, from perivascular precursors expressing Pdgfrb. Pdgfrb encodes the platelet-derived growth factor receptor β chain (Pdgfrβ protein). The highly adipogenic subpopulation of Pdgfrβ+ cells can be identified by the expression of Pparg, the master regulatory gene of adipocyte differentiation, as well as its upstream regulatory factor, Zfp423[35–37]. These Pparg/Zfp423-expressing Pdgfrβ+ cells express several mural cell (pericyte/smooth muscle) markers and reside directly adjacent to the endothelium in WAT blood vessels. Here, we utilized inducible genetic models to manipulate the expression of Pparg in Pdgfrb-expressing perivascular precursors of adult mice. Given the well-established role of Pparγ in adipogenesis, these models allowed us to investigate the consequences of manipulating the adipogenic capacity of these precursors in adult animals. Doxycycline-inducible gain- and loss of function studies in mice provide evidence that de novo visceral adipocyte differentiation in the face of caloric excess protects against unhealthy WAT remodeling in obesity, and is associated with preservation of the insulin-sensitizing hormone adiponectin and systemic insulin sensitivity. In line with these observations, we found that the ability of TZDs to trigger healthy WAT remodeling is dependent on mural cell Pparg. These data highlight the protective effects of de novo adipocyte differentiation in adult mice, even in visceral WAT depots, and point to Pdgfrβ+ adipocyte precursors as potential cellular targets for therapeutic intervention in type 2 diabetes.

## Results

**Adipocytes emerge from Pdgfrβ+ precursors in select WAT depots.** WAT expansion associated with HFD feeding in male C57BL/6 mice occurs in a depot-specific manner. In response to HFD feeding, the subcutaneous inguinal WAT (iWAT) depot expands almost exclusively by adipocyte hypertrophy, despite the abundance of adipose progenitors in this tissue[38,39]. The gonadal (epididymal) WAT depot, most often used as the representative visceral WAT depot, expands through both adipocyte hypertrophy and adipocyte hyperplasia. We have previously utilized a doxycycline (Dox)-inducible lineage-tracing system (Pdgfrb[rtTA]; TRE-Cre; Rosa26R[mT/mG]; herein, "MuralChaser mice") (Supplementary Fig. 1a) to follow the fate of Pdgfrb-expressing cells in mice under various conditions that promote adipose remodeling[37]. We uncovered adipose Pdgfrβ+ mural cells as developmental precursors of visceral white adipocytes formed in association with 8 weeks of high-fat diet feeding. Importantly, Pdgfrβ+ precursors appeared to contribute little to the homeostatic control of adipocyte number under chow-fed conditions.

Here, we have extended this pulse-chase lineage-tracing approach to analyze other intra-abdominal depots of diet-induced obese (DIO) mice (Supplementary Fig. 1b). Following 16 weeks of HFD feeding (60% kcal from fat), newly formed adipocytes emerging from the Pdgfrb-lineage can also be readily identified (labeled as mGFP+ adipocytes) in the perirenal depot (pWAT), at a frequency similar to that observed in the gonadal WAT (gWAT) depot (Supplementary Fig. 1c–j, w). Consistent with our prior study, very few mGFP+ cells were observed in iWAT depots (Supplementary Fig. 1k–p, w). This is perhaps not surprising given the relatively low rate of adipogenesis that naturally occurs in this depot during HFD feeding. Interestingly,

only occasional mGFP+ adipocytes were observed in the mesenteric WAT (mWAT) depot, suggesting that mWAT expansion in DIO mice is largely independent of Pdgfrβ+ mural adipose precursors (Supplementary Figure 1q–v, w). Altogether, these data highlight the selective contribution of *Pdgfrb*-expressing precursors to adipocyte hyperplasia in vivo.

**Mural *Pparg* overexpression drives healthy visceral WAT expansion in obesity.** Pparγ is both necessary and sufficient for adipocyte differentiation[40–43]. *Pparg* expression is a defining feature of highly committed adipocyte precursors, and *Pparg*-expressing Pdgfrβ+ mural cells appear predominantly within adipose tissues of adult mice[36]. Manipulation of *Pparg* levels

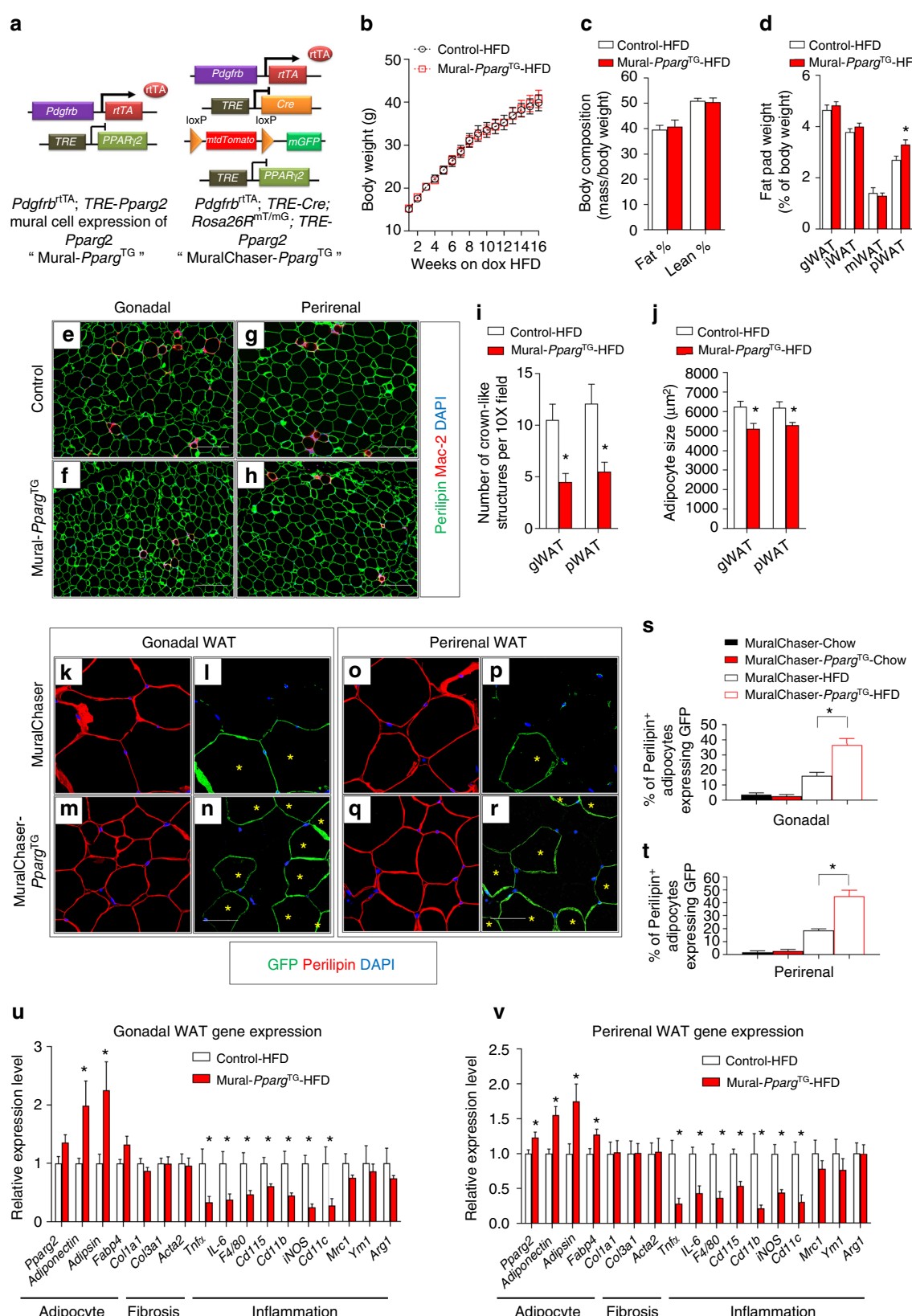

dramatically alters the adipogenic capacity of precursors[42]. This affords the opportunity to define the importance of the adipogenic capacity of Pdgfrβ+ precursors to adipose tissue homeostasis in adults. We first asked whether increasing the adipogenic capacity of Pdgfrb-expressing precursors is sufficient to promote healthy WAT remodeling. We derived transgenic animals expressing Pparg2 in Pdgfrb-expressing cells in a Dox-dependent manner (Pdgfrb^rtTA; TRE-Pparg2; herein, "Mural-Pparg^TG") (Fig. 1a). The addition of doxycycline to purified Pdgfrβ+ cells from Mural-Pparg^TG animals leads to transgenic expression of FLAG-tagged Pparγ2 and improves the adipogenic capacity of these cells in vitro (Supplementary Fig. 2a, b). Seven days of doxycyline treatment of Mural-Pparg^TG mice in vivo leads to Pparγ2 protein expression in the stromal fraction of all adipose depots examined (Supplementary Fig. 2c). Importantly, transgene expression is not detected in freshly isolated mature adipocytes, consistent with lack of Pdgfrb expression in differentiated fat cells (Supplementary Fig. 2c, d). We also assessed how transgene expression impacts overall mRNA levels of Pparg2 in isolated Pdgfrβ+ cells. Pparg2 levels are very low and barely detectable by qPCR in Pdgfrβ+ cells from control mice; however, upon transgene activation, overall levels of Pparg2 increase to ~25% of levels found in mature adipocytes. This occurs in both gWAT and iWAT depots. Consistent with the western blot analysis and restricted pattern of Pdgfrb expression, overall levels of Pparg2 in mature adipocytes are not impacted in this model (Supplementary Fig. 2d). As a result, this model enables us to assess the consequences of increasing the adipogenic capacity of mural cells, without directly manipulating the expression of genes in mature adipocytes. In parallel, we also introduced TRE-Cre and Rosa26R^mT/mG alleles into the Mural-Pparg^TG strain (Fig. 1a). This derivative model (Pdgfrb^rtTA; TRE-Cre; TRE-Pparg2; Rosa26R^mT/mG; herein "MuralChaser-Pparg^TG") enables lineage-tracking of Pdgfrβ+ cells overexpressing Pparg2, and thus visualization of any newly formed adipocytes.

We first administered control (Pdgfrb^rtTA or TRE-Pparg2 single transgenics) and Mural-Pparg^TG mice a Dox-containing chow diet for 16 weeks starting from 6 weeks of age (Supplementary Fig. 3a). During the 16-week period, body weights and overall adiposity of Mural-Pparg^TG mice remained indistinguishable from control animals (Supplementary Fig. 3b, c). We did not detect any obvious histological differences in adipocyte size or overall WAT morphology between control and transgenic

animals (Supplementary Fig. 3d–l). In particular, mRNA levels of adipocyte-, vascular-, and inflammatory-related genes in WAT depots of Mural-Pparg^TG mice were comparable to corresponding levels found in WAT of control animals (Supplementary Fig. 3m–o). We analyzed mGFP reporter expression in parallel cohorts of control (MuralChaser mice) and MuralChaser-Pparg^TG mice maintained on Dox-containing chow diet for the same period of time. Surprisingly, Pparg2 overexpression did not drive the formation of new adipocytes from Pdgfrb-expressing cells in any of the WAT depots examined (Supplementary Fig. 3p–ae). Thus, transgenic expression of Pparg2 alone, at least to the level achieved here, was insufficient to trigger adipocyte differentiation from Pdgfrβ+ precursors under these conditions (chow diet).

We also challenged control and Mural-Pparg^TG animals to HFD feeding for a 16-week period. During the course of HFD feeding, body weights and overall adiposity of Mural-Pparg^TG mice remained indistinguishable from control animals (Fig. 1b, c). The weights of most individual adipose depots from obese Mural-Pparg^TG mice were similar to those in control animals (Fig. 1d). The lone exception was perirenal WAT; this depot was mildly but significantly larger in the HFD-fed Mural-Pparg^TG animals (Fig. 1d). Histological analysis of adipose depots from obese control and Mural-Pparg^TG mice revealed a striking difference in tissue cellularity (Fig. 1e–j). Despite being equal or larger in mass, gWAT and pWAT from Mural-Pparg^TG mice contained significantly smaller adipocytes than those found in corresponding depots from control animals (Fig. 1j). This phenotype coincided with lower tissue mRNA levels of several pro-inflammatory genes and the overall presence of crown-like structures (Fig. 1i, u, v). In parallel cohorts of HFD-fed MuralChaser-Pparg^TG mice, we quantified the occurrence of mGFP+ adipocytes. After 16 weeks of HFD, ~20% of adipocytes found in gWAT and pWAT of MuralChaser mice represent newly formed adipocytes that emerge from mural progenitors. This number nearly doubles in those depots in HFD-fed MuralChaser-Pparg^TG (Fig. 1k–t). Taken together, these data indicate that driving adipocyte hyperplasia through differentiation from mural progenitors does not lead to an increase in overall body weight or adiposity under obesogenic conditions, but rather leads to healthy visceral WAT remodeling.

In MuralChaser-Pparg^TG mice, Pparg2 expression is induced in Pdgfrβ+ mural cells of all adipose depots; however, we did not

---

**Fig. 1** Mural Pparg overexpression drives healthy visceral WAT expansion in obesity. **a** Pdgfrb^rtTA; TRE-Pparg2 (Mural-Pparg^TG) mice are generated by breeding the Pdgfrb^rtTA transgenic mice to animals expressing Pparg2 under the control of the tet-response element (TRE-Pparg2). Littermates carrying only Pdgfrb^rtTA or TRE-Pparg2 alleles were used as the control animals for Mural-Pparg^TG. Pdgfrb^rtTA; TRE-Cre; Rosa26R^mT/mG; TRE-Pparg2 (MuralChaser-Pparg^TG) mice are generated by breeding the "MuralChaser" (Pdgfrb^rtTA; TRE-Cre; Rosa26R^mT/mG) mice to animals carrying TRE-Pparg2 transgene. MuralChaser mice were used as control animals for MuralChaser-Pparg^TG. **b** Control and Mural-Pparg^TG mice were fed a standard chow diet until 6 weeks of age before being switched to dox-containing high-fat diet (HFD). Body weights were measured weekly following the onset of HFD feeding. Control-HFD, n = 8; Mural-Pparg^TG-HFD, n = 9. Data points represent mean + s.e.m. **c** Fat mass and lean mass (normalized to body weight) of control and Mural-Pparg^TG mice after 16 weeks of dox-HFD feeding. Control-HFD, n = 8; Mural-Pparg^TG-HFD, n = 9. Bars represent mean + s.e.m. **d** Fat pad weight (normalized to body weight) of control and Mural-Pparg^TG mice after 16 weeks of dox-HFD feeding. * denotes p < 0.05 from Student's t-test. Control-HFD, n = 8; Mural-Pparg^TG-HFD, n = 9. Bars represent mean + s.e.m. **e–h** Representative immunofluorescence staining of Perilipin (green) and Mac-2 (red) in **e**, **f** gonadal and **g**, **h** perirenal WAT paraffin sections obtained from control and Mural-Pparg^TG mice after 16 weeks of dox-HFD feeding. Scale bar, 200 μm. **i** Number of crown-like structures (Mac-2 positive) in the indicated depots from control and Mural-Pparg^TG mice after 16 weeks of dox-HFD feeding. * denotes p < 0.05 from Welch's t-test. n = 24 randomly chosen ×10 magnification fields from six individual animals. Bars represent mean + s.e.m. **j** Average adipocyte size in indicated fat depots from control and Mural-Pparg^TG mice after 16 weeks of dox-HFD feeding. * denotes p < 0.05 from Student's t-test. n = 6 per genotype. Bars represent mean + s.e.m. **k–r** Representative ×63 magnification confocal immunofluorescence images of **k–n** gonadal and **o–r** perirenal WAT sections from MuralChaser and MuralChaser-Pparg^TG mice after 16 weeks dox-diet feeding. Sections were stained with anti-GFP (green) and anti-Perilipin (red) antibodies and counterstained with DAPI (blue; nuclei). * indicates GFP-labeled perilipin-positive cells. Scale bar, 50 μm. **s–t** Percentage of perilipin-positive adipocytes expressing GFP in **s** gonadal and **t** perirenal WAT from MuralChaser and MuralChaser-Pparg^TG mice maintained on dox-diets for 16 weeks. Two-way ANOVA, *p < 0.05; n = 6 individual depots per group. Bars represent mean + s.e.m. **u**, **v** Relative mRNA levels of indicated genes in **u** gonadal and **v** perirenal WAT obtained from control and Mural-Pparg^TG mice after 16 weeks of dox-HFD feeding. * denotes p < 0.05 from Welch's t-test. n = 6 per genotype. Bars represent mean + s.e.m.

observe a significant degree of adipocyte differentiation from mural adipocyte progenitors within iWAT or mWAT of HFD-fed animals (Supplementary Fig. 4a–h). This result is surprising given the aforementioned in vitro data demonstrating the ease in which transgenic *Pparg* drives adipocyte differentiation from iWAT stromal cells in vitro. Jeffery et al.[44] recently described the importance of the local adipose tissue microenvironment in regulating adipocyte progenitor activity. Thus, our data may suggest that strong suppressive signals exist within the local iWAT and mWAT microenvironment that inhibit adipogenesis from these mural cells, even when Pparγ is overexpressed. Moreover, we did not observe any histological abnormalities or changes in adipocyte size in these depots (Supplementary Fig. 4i–n), and the expression of adipocyte-, fibrosis-, and inflammation-related genes appeared only marginally affected (Supplementary Fig. 4o). Thus, the hyperplastic WAT phenotype in MuralChaser-*Pparg*[TG] animals is largely confined to select visceral adipose depots.

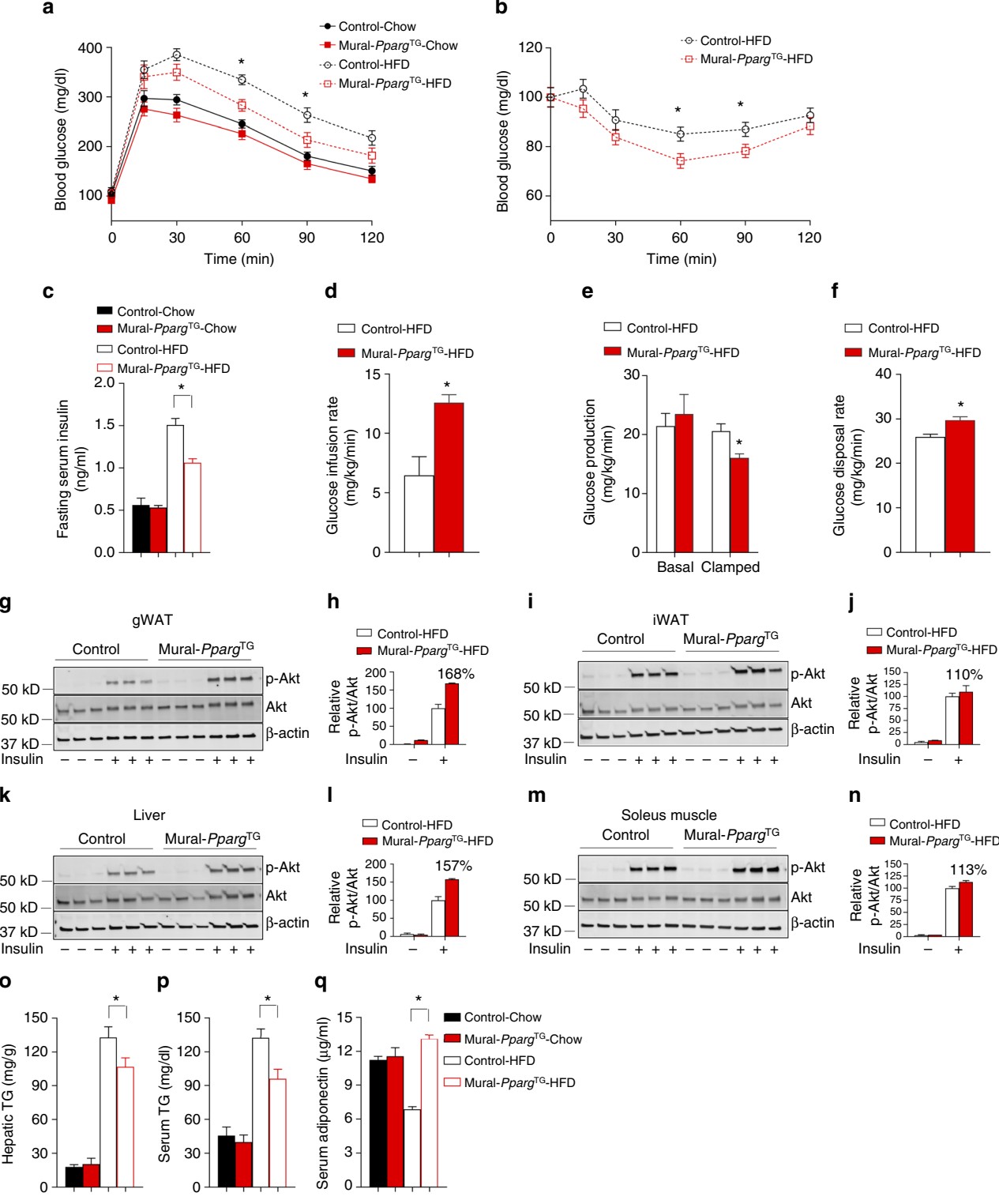

**Hyperplastic visceral WAT expansion is associated with improved nutrient homeostasis.** This unique model of healthy visceral WAT remodeling allows us to ask whether hyperplastic visceral WAT expansion in obesity is linked to either improved or impaired nutrient homeostasis. Indeed, obese Mural-$Pparg^{TG}$ animals were more glucose tolerant than obese control mice (Fig. 2a). Serum insulin levels and insulin tolerance tests suggested that Mural-$Pparg^{TG}$ animals were more insulin sensitive (Fig. 2b, c). We further explored insulin sensitivity in these animals by performing hypersulinemic–euglycemic clamp assays. The glucose infusion rate needed to maintain euglycemia (~137 mg/dl) was higher in obese Mural-$Pparg^{TG}$ animals as compared to obese control animals (Fig. 2d). This difference could not be explained by minimal differences in glucose or insulin levels achieved during the clamped state. This demonstrates an increase in whole-body insulin sensitivity, consistent with the aforementioned insulin tolerance tests. Endogenous glucose output was suppressed more efficiently in Mural-$Pparg^{TG}$ mice during the clamped state, likely reflecting improved insulin sensitivity at the level of the liver (Fig. 2e). $^3$H-glucose kinetics revealed that the rate of whole-body glucose disposal was elevated significantly in Mural-$Pparg^{TG}$ mice (Fig. 2f). To complement these studies, we also assessed the response of individual tissues to the actions of insulin, as reflected by the phosphorylation of its downstream signal transducer, Akt (pAKT). Levels of pAKT were significantly higher in the gWAT and liver of Mural-$Pparg^{TG}$ mice following insulin stimulation (Fig. 2g, h, k, l). Levels of pAKT in the iWAT and soleus muscle of these animals were comparable to levels found in corresponding tissues from control mice (Fig. 2i, j, m, n). The improved insulin sensitivity of the liver was associated with moderate improvements in lipid homeostasis. Notably, serum and hepatic triglycerides were lower in Mural-$Pparg^{TG}$ mice when compared to control animals (Fig. 2o, p). Remarkably, serum levels of adiponectin were dramatically elevated in obese Mural-$Pparg^{TG}$ mice, reaching levels normally observed in lean animals (Fig. 2q). All together, these data reveal that obese Mural-$Pparg^{TG}$ mice exhibit a local adipose and systemic phenotype reminiscent of the "insulin-sensitive" obese.

One possible mechanism leading to sustained insulin sensitivity in obese Mural-$Pparg^{TG}$ mice is the remarkable preservation of serum levels of adiponectin. Adiponectin is a well-described physiological regulator of insulin sensitivity[45]. We asked whether the local and systemic phenotypes observed in Mural-$Pparg^{TG}$ mice were dependent on the presence of adiponectin. We bred Mural-$Pparg^{TG}$ mice to animals carrying inactive *Adipoq* (Adiponectin) alleles (Mural-$Pparg^{TG}$; $Adiponectin^{-/-}$ mice)

and reassessed the impact of mural *Pparg* overexpression on adipose remodeling and systemic metabolic health following HFD feeding. Body weights, overall adiposity, and the mass of individual fat depots were comparable between control (*Adiponectin$^{-/-}$* mice) and Mural-$Pparg^{TG}$; $Adiponectin^{-/-}$ mice (Fig. 3a–c). However, the average cell size and the degree of macrophage accumulation was lower in the gWAT and pWAT depots of Mural-$Pparg^{TG}$; $Adiponectin^{-/-}$ mice (Fig. 3d–n; Supplementary Fig. 5a–c). These data indicate that the beneficial effects on local WAT remodeling elicited by mural *Pparg* expression are not fully dependent on adiponectin. However, in the absence of *adiponectin*, glucose tolerance and insulin sensitivity are no longer significantly improved by mural Pparg overexpression (Fig. 3o–r). Thus, the systemic improvements in glucose homeostasis observed in Mural-$Pparg^{TG}$ mice appear adiponectin-dependent. This suggests that the healthy visceral WAT in these animals contributes, at least in part, to the improved metabolic phenotype in this model.

**Mural *Pparg* is dispensable for WAT homeostasis in lean adult mice.** Lineage tracing using the MuralChaser model highlights a natural contribution of *Pdgfrb*-expressing precursors to adipocyte hyperplasia in vivo; however, whether the adipogenic capacity of these cells is essential for healthy WAT remodeling and expansion in different settings has been unclear. To address this question, we generated a mouse model in which *Pparg* is inactivated in *Pdgfrb*-expressing mural cells in a Dox-inducible manner (*Pdgfrb*$^{rtTA}$; *TRE-Cre*, *Pparg*$^{loxp/loxp}$; denoted herein as Mural-$Pparg^{KO}$ mice) (Fig. 4a). Administration of Dox-containing chow diet to Mural-$Pparg^{KO}$ mice leads to efficient inactivation of *Pparg*; *Pparg* mRNA levels are reduced by 90% within purified Pdgfrβ$^+$ cells (Supplementary Fig. 6a). Accordingly, Pdgfrβ$^+$ mural cells isolated from Mural-$Pparg^{KO}$ mice lacked the ability to undergo adipogenesis in vitro (Supplementary Fig. 6b). The *Pdgfrb* promoter is not active in mature adipocytes; therefore, *Pparg* is not inactivated in existing mature adipocytes in this model (Supplementary Fig. 6c).

We first utilized this model to evaluate whether mural cell *Pparg* is important for the adipose tissue homeostasis in lean (chow diet fed) adult mice. Control (*Pdgfrb*$^{rtTA}$; *Pparg*$^{loxp/loxp}$) and Mural-$Pparg^{KO}$ mice were administered Dox-containing chow diet for up to 20 weeks (Supplementary Fig. 6d). We did not observe any significant differences in body weight or body composition between chow-fed control and Mural-$Pparg^{KO}$ mice (Supplementary Fig. 6e, f). Histological and gene expression

---

**Fig. 2** Mural *Pparg* overexpression leads to metabolic benefits in obesity. **a** Glucose tolerance tests of control and Mural-$Pparg^{TG}$ mice after 16 weeks of dox-diet feeding. Two-way ANOVA, *$p < 0.05$; Control-Chow, $n = 7$; Mural-$Pparg^{TG}$-Chow, $n = 7$; Control-HFD, $n = 8$; Mural-$Pparg^{TG}$-HFD, $n = 9$. Data points represent mean + s.e.m. **b** Insulin tolerance tests of control and Mural-$Pparg^{TG}$ mice after 16 weeks of dox-HFD feeding. Student's t-test, *$p < 0.05$; Control-HFD, $n = 8$; Mural-$Pparg^{TG}$-HFD, $n = 9$. Data points represent mean + s.e.m. **c** Six-hour fasting serum insulin levels in control and Mural-$Pparg^{TG}$ mice after 16 weeks of dox-diet feeding. Two-way ANOVA, *$p < 0.05$; Control-Chow, $n = 7$; Mural-$Pparg^{TG}$-Chow, $n = 7$; Control-HFD, $n = 8$; Mural-$Pparg^{TG}$-HFD, $n = 9$. Bars represent mean + s.e.m. **d** The glucose infusion rate needed to maintain euglycemia (~137 mg/dl) during hyperinsulinemic–euglycemic clamp assays of conscious unrestrained control and Mural-$Pparg^{TG}$ mice after 16 weeks of dox-HFD feeding. Student's t-test, *$p < 0.05$; Control-HFD, $n = 7$; Mural-$Pparg^{TG}$-HFD, $n = 5$. Bars represent mean + s.e.m. **e** Basal and clamped rates of endogenous glucose production during hyperinsulinemic–euglycemic clamp assays of control and Mural-$Pparg^{TG}$ mice after 16 weeks of dox-HFD feeding. Student's t-test, *$p < 0.05$; Control-HFD, $n = 7$; Mural-$Pparg^{TG}$-HFD, $n = 5$. Bars represent mean + s.e.m. **f** Rates of glucose disposal during hyperinsulinemic–euglycemic clamp assays of control and Mural-$Pparg^{TG}$ mice after 16 weeks of dox-HFD feeding. Student's t-test, *$p < 0.05$; Control-HFD, $n = 7$; Mural-$Pparg^{TG}$ -HFD, $n = 5$. Bars represent mean + s.e.m. **g–n** Western blot analysis of phosphorylated Akt (pAkt), total Akt, and β-actin protein levels in tissue extracts of **g**, **h** gonadal WAT, **i**, **j** inguinal WAT, **k–l** liver, and **m**, **n** soleus muscle from control and Mural-$Pparg^{TG}$ mice after 16 weeks of dox-HFD feeding. For quantification, intensity of pAkt band is normalized to that of total Akt band. The mean value of pAkt/Akt intensity ratio of insulin-treated control mice samples were set as 100%. $n = 3$ individual mice per genotype. Bars represent mean + s.e.m. **o** Hepatic and **p** serum triglycerides levels in control and Mural-$Pparg^{TG}$ mice after 16 weeks of dox-diet feeding. Two-way ANOVA, *$p < 0.05$; Control-Chow, $n = 7$; Mural-$Pparg^{TG}$-Chow, $n = 7$; Control-HFD, $n = 8$; Mural-$Pparg^{TG}$-HFD, $n = 9$. Bars represent mean + s.e.m. **q** Serum adiponectin levels in control and Mural-$Pparg^{TG}$ mice after 16 weeks of dox-diet feeding. Two-way ANOVA, *$p < 0.05$; Control-Chow, $n = 7$; Mural-$Pparg^{TG}$-Chow, $n = 7$; Control-HFD, $n = 8$; Mural-$Pparg^{TG}$ -HFD, $n = 9$. Bars represent mean + s.e.m.

analyses of adipose depots suggested that mural cell *Pparg* was dispensable for maintenance of WAT mass in lean mice (Supplementary Fig. 6g); adipose tissue morphology and the expression of adipocyte-, vascular-, and inflammatory-related genes were not impacted by mural cell *Pparg* deficiency (Supplementary Fig. 6h–s). Moreover, Mural-*Pparg*KO mice

maintained on chow diet did not exhibit any impairment in glucose tolerance Supplementary Fig. 6t). These data stand in contrast to the reported consequences of ablating *Pparg* in the smooth muscle actin (SMA) lineage; inducible deletion of Pparg in the SMA+ lineage (*Acta2*-CreERT2) of weaning age mice results in a progressive deterioration of adipose tissue mass[46].

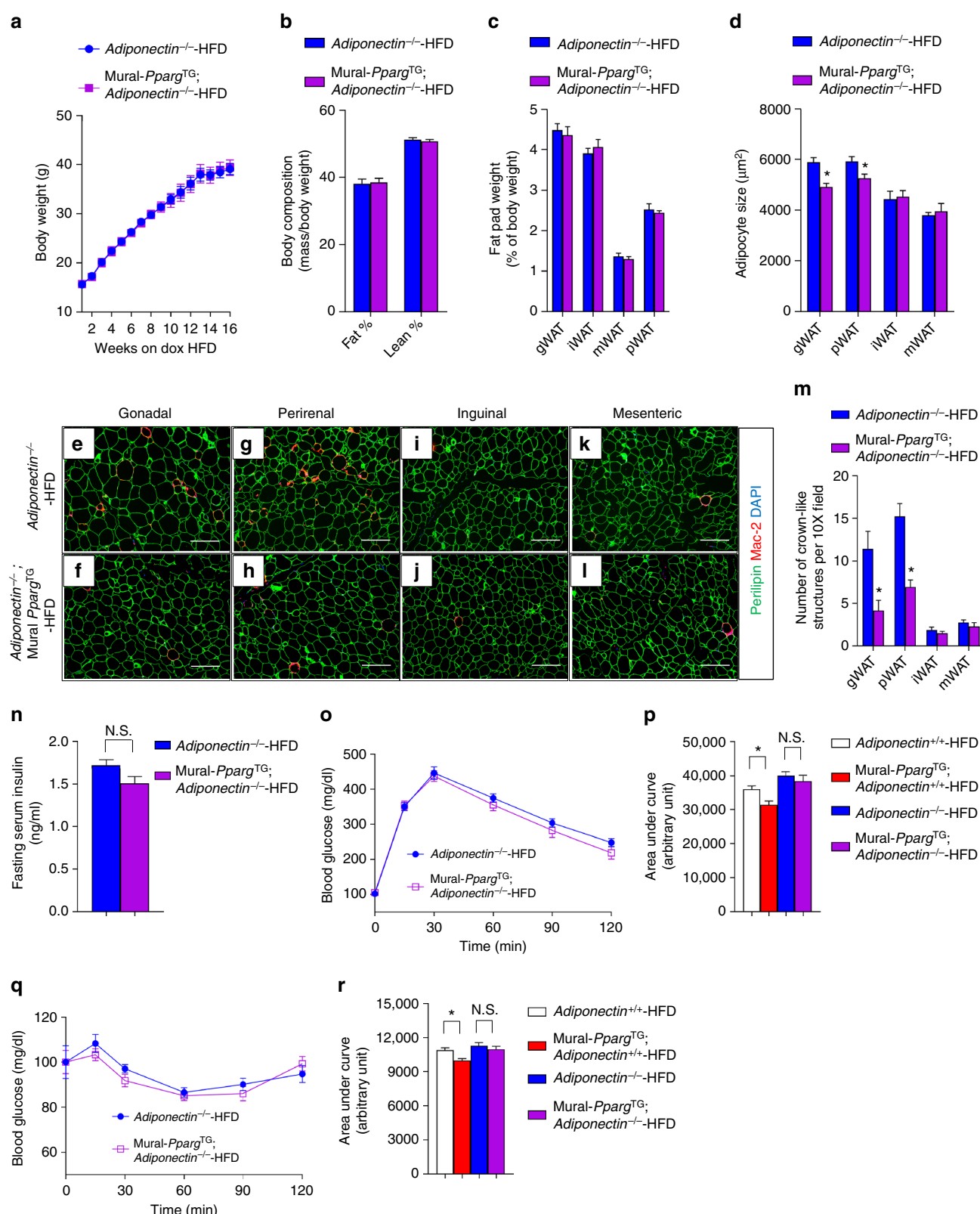

This discrepancy may reflect differential targeting of adipose tissue perivascular cell populations. *Acta2*-CreERT2 may targets additional perivascular cells, including those playing a role in the homeostatic maintenance of adipose tissue mass under these conditions.

**Mural *Pparg* is critical for healthy visceral WAT expansion in obesity**. We next administered control and Mural-*Pparg*$^{KO}$ mice HFD for up to 20 weeks. During the course of HFD feeding, the body weights and overall adiposity of Mural-*Pparg*$^{KO}$ mice again remained indistinguishable from controls animals (Fig. 4b, c). However, by 20 weeks of HFD feeding, Mural-*Pparg*$^{KO}$ mice exhibit a shift in body fat distribution (Fig. 4d). The mass of pWAT and gWAT depots of Mural-*Pparg*$^{KO}$ mice was significantly smaller than corresponding depots from control animals (Fig. 4d). Notably, this phenotype correlates with the major sites of mural cell-derived adipogenesis in DIO mice (see Supplementary Fig. 1). The iWAT was slightly, but significantly, larger in the Mural-*Pparg*$^{KO}$ mice than in controls (Fig. 4d). This may reflect an attempt to compensate for the smaller visceral depots in these animals. Histological analysis revealed a striking phenotype specific to gWAT and pWAT of Mural-*Pparg*$^{KO}$ mice. These visceral depots exhibit robust macrophage infiltration (Fig. 4e–h, m) and increased collagen deposition (Fig. 4i–l). These are two signature hallmarks of pathological adipose expansion. The average size of perirenal adipocytes was significantly larger in Mural-*Pparg*$^{KO}$ mice, indicating increased adipocyte hypertrophy (Fig. 4n). Gene expression analysis revealed lower mRNA levels of adipocyte-selective genes in both visceral depots of Mural-*Pparg*$^{KO}$ mice, while the expression of genes related to fibrosis and inflammation were elevated (Fig. 4o, p). Overall, the relative abundance of Pdgfrβ$^+$ cells within the adipose stromal compartment did not appear to be altered by *Pparg* deficiency (Supplementary Fig. 7a, b); however, levels of collagens and other fibrosis markers were elevated within purified Pdgfrβ$^+$ cells from gWAT and pWAT (Supplementary Fig. 7c). This suggests that *Pparg*-deficient mural cells triggered to undergo adipogenesis may instead adopt a myofibroblast phenotype. Indeed, TGFβ-stimulated induction of myofibroblast-related transcripts occurred more robustly in purified Pdgfrβ$^+$ cells lacking *Pparg* than in control cells (Supplementary Fig. 7d, e). Importantly, these adipose and mural cell phenotypes were restricted in vivo to those depots where mural adipose precursors give rise to adipocytes

upon HFD feeding. Despite efficient deletion in mural cells of iWAT or mWAT, we did not observe histological abnormalities (Supplementary Fig. 8a–j) or robust gene expression changes in isolated mural cells of whole tissue of obese Mural-*Pparg*$^{KO}$ mice (Supplementary Fig. 8k–n). This suggests that the pathological remodeling of visceral WAT depots observed in Mural-*Pparg*$^{KO}$ animals is triggered when mural cells lose their adipogenic capacity, rather than the loss of *Pparg* in mural cells per se.

Clinically, pathological WAT expansion typically correlates with features of metabolic syndrome, including impaired glucose and lipid homeostasis. After 10 weeks of HFD feeding, obese Mural-*Pparg*$^{KO}$ mice performed similarly to control animals in glucose tolerance tests (Supplementary Fig. 9a). Moreover, serum and hepatic triglyceride levels were not significantly different (Supplementary Fig. 9b, c). However, by this stage, the visceral WAT appeared relatively less responsive to the actions of insulin, as reflected by levels of pAKT. As compared to control animals, insulin-induced Akt phosphorylation was significantly impaired in the gWAT, but not in iWAT or liver, of Mural-*Pparg*$^{KO}$ mice (Supplementary Fig. 9d–f). Levels of pAKT were moderately reduced (average reduction ~14%) in the soleus muscle (Supplementary Fig. 9g). This local visceral WAT insulin resistance correlates with the abnormal histological phenotype of this depot in these animals. By 20 weeks of HFD feeding, obese Mural-*Pparg*$^{KO}$ mice became relatively more glucose intolerant and insulin resistant than obese controls (Supplementary Fig. 9h–j). After this prolonged HFD feeding period, levels of phosphorylated AKT were significantly lower in all examined tissues of the obese Mural-*Pparg*$^{KO}$ mice when compared to corresponding levels in controls; however, the most dramatic effects remained in the gWAT (Supplementary Fig. 9k–n). Moreover, levels of hepatic and serum triglycerides were relatively higher in obese Mural-*Pparg*$^{KO}$ mice (Supplementary Fig. 9o, p). Circulating levels of adiponectin are typically reduced as animals become obese; levels of adiponectin fell even further in Mural-*Pparg*$^{KO}$ mice (Supplementary Fig. 9q). Taken all together, these data highlight the importance of mural cell *Pparg* in the prevention of maladaptive visceral WAT remodeling associated with HFD feeding, and further links visceral WAT health to the development of systemic insulin sensitivity.

**Mural cell *Pparg* is required for TZD-induced visceral WAT remodeling**. To date, the exact physiological signals that trigger

**Fig. 3** The metabolic benefits of mural *Pparg* overexpression depend on adiponectin. **a** *Adiponectin*-deficient (*Adiponectin*$^{-/-}$) mice and *Adiponectin*$^{-/-}$; *Pdgfrb*$^{rtTA}$; *TRE-Pparg2* (*Adiponectin*$^{-/-}$; Mural-*Pparg*$^{TG}$) mice were fed a standard chow diet until 6 weeks of age before being switched to dox-containing HFD. Body weights were measured weekly following the onset of HFD feeding. $n = 6$ per genotype. Data points represent mean + s.e.m. **b** Fat mass and lean mass (normalized to body weight) of *Adiponectin*$^{-/-}$ and *Adiponectin*$^{-/-}$; Mural-*Pparg*$^{TG}$ mice after 16 weeks of dox-HFD feeding. $n = 6$ per genotype. Bars represent mean + s.e.m. **c** Average adipocyte size in indicated fat depots from control and Mural-*Pparg*$^{TG}$ mice after 16 weeks of dox-diet feeding. Bars represent mean + s.e.m. **d** Fat pad weight (normalized to body weight) of *Adiponectin*$^{-/-}$ and *Adiponectin*$^{-/-}$; Mural-*Pparg*$^{TG}$ mice after 16 weeks of dox-HFD feeding. * denotes $p < 0.05$ from Student's *t*-test. $n = 6$ per genotype. Bars represent mean + s.e.m. **e–l** Representative immunofluorescence staining of Perilipin (green) and Mac-2 (red) in **e**, **f** gonadal, **g**, **h** perirenal WAT, **i**, **j** inguinal WAT, and **k–l** mesenteric WAT paraffin sections obtained from *Adiponectin*$^{-/-}$ and *Adiponectin*$^{-/-}$; Mural-*Pparg*$^{TG}$ mice after 16 weeks of dox-HFD feeding. Scale bar, 200 μm. **m** Numbers of crown-like structures (Mac-2 positive) in the indicated depots from *Adiponectin*$^{-/-}$ and *Adiponectin*$^{-/-}$; Mural-*Pparg*$^{TG}$ mice after 16 weeks of dox-HFD feeding. * denotes $p < 0.05$ from Welch's *t*-test. $n = 24$ randomly chosen ×10 magnification fields from six individual animals. Bars represent mean + s.e.m. **n** Six-hour fasting serum insulin levels in *Adiponectin*$^{-/-}$ and *Adiponectin*$^{-/-}$; Mural-*Pparg*$^{TG}$ mice after 16 weeks of dox-diet feeding. n.s. denotes not statistically significant. $n = 6$ per genotype. Bars represent mean + s.e.m. **o** Glucose tolerance tests of *Adiponectin*$^{-/-}$ and *Adiponectin*$^{-/-}$; Mural-*Pparg*$^{TG}$ mice after 16 weeks of dox-diet feeding. $n = 6$ per genotype. Data points represent mean + s.e.m. **p** Glucose area under the curve measurements during glucose tolerance test from mice of indicated genotypes after 16 weeks of dox-diet feeding. Two-way ANOVA, *$p < 0.05$; n.s. denotes not statistically significant; *Adiponectin*$^{+/+}$, $n = 8$; *Adiponectin*$^{+/+}$; Mural-*Pparg*$^{TG}$, $n = 9$; *Adiponectin*$^{-/-}$, $n = 6$; *Adiponectin*$^{-/-}$; Mural-*Pparg*$^{TG}$, $n = 6$. Bars represent mean + s.e.m. **q** Insulin tolerance test of *Adiponectin*$^{-/-}$ and *Adiponectin*$^{-/-}$; Mural-*Pparg*$^{TG}$ mice after 16 weeks of dox-diet feeding. $n = 6$ per genotype. Data points represent mean + s.e.m. **r** Glucose area under the curve measurements during insulin tolerance test from mice of indicated genotypes after 16 weeks of dox-diet feeding. Two-way ANOVA, *$p < 0.05$; n.s. denotes not statistically significant; *Adiponectin*$^{+/+}$, $n = 8$; *Adiponectin*$^{+/+}$; Mural-*Pparg*$^{TG}$, $n = 9$; *Adiponectin*$^{-/-}$, $n = 6$; *Adiponectin*$^{-/-}$; Mural-*Pparg*$^{TG}$, $n = 6$. Bars represent mean + s.e.m.

adipocyte hyperplasia in the setting of caloric excess remain unclear; however, it is well documented that anti-diabetic Pparγ agonists (TZDs) trigger a healthy hyperplastic expansion of adipose tissue[47,48]. In patients treated long-term with TZDs, there is a noticeable expansion of subcutaneous adipose tissue, often at the expense of visceral WAT[49]. This healthy body fat distribution is believed to be a critical component of how TZDs improve nutrient homeostasis. It is increasingly apparent that TZDs target Pparγ in multiple cells types, including immune cells, adipocytes, preadipocytes, perivascular cells, hepatocytes, and skeletal myocytes[50–58]. In obese rats, short-term TZD treatment for 4 weeks resulted in both visceral and subcutaneous WAT remodeling, characterized by the appearance of relatively smaller adipocytes and a reduction in tissue inflammation. Thus, we asked whether TZDs trigger adipocyte hyperplasia in these depots through differentiation from *Pdgfrb*-expressing adipose progenitors. MuralChaser mice were first rendered obese by 10 weeks of HFD feeding (Fig. 5a). We then induced mGFP reporter expression in mural cells of obese mice by feeding Dox-containing HFD for 1 week. Rosiglitazone (Rosi) was

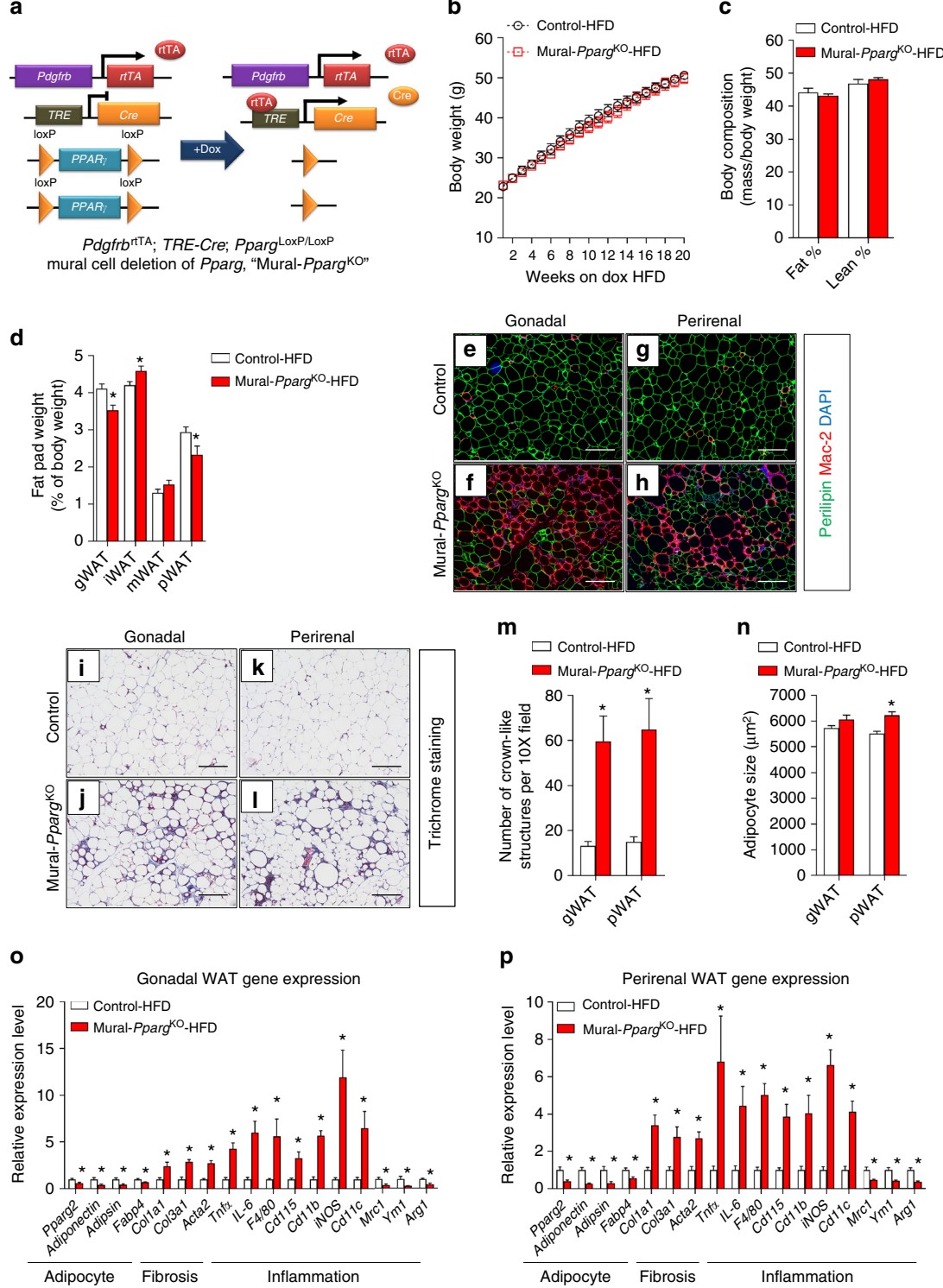

subsequently administered daily by gavage for four continuous weeks while maintaining animals on HFD devoid of doxycycline (Fig. 5a). After the 4-week treatment period, the mass of all depots examined increased and mGFP+ adipocytes were present in all depots, albeit to a much greater degree in pWAT and gWAT than in iWAT (Fig. 5b–o). This correlated with a reduction in adipocyte size in these depots (Fig. 5p–u). Thus, under these conditions, TZDs trigger significant visceral adipocyte hyperplasia in obese mice, at least in part, by triggering adipocyte differentiation from mural adipocyte progenitors.

These lineage-tracing data raise the question as to whether some of the beneficial effects of TZDs on adipose tissue depend on the adipogenic capacity of *Pdgfrb*-expressing precursors. We tested this hypothesis by treating obese control and Mural-*Pparg*^KO animals with Rosi shortly after inactivating *Pparg* in mural cells (Supplementary Fig. 10a). Rosi treatment impacted obese control animals in a manner consistent with prior studies[59,60]. When compared to vehicle-treated animals, control animals on Rosi gained body weight and adiposity (Fig. 6a–c). The mass of all subcutaneous and visceral adipose depots examined increased (Fig. 6d); however, adipocytes tended to be smaller in size and fewer crown-like structures were evident (Fig. 6e, g, i, k, m–p). Mural cell inactivation of *Pparg* did not significantly impact the ability of Rosi to trigger weight gain and increase adiposity (Fig. 6a–c). Moreover, adipose depot mass was unaffected by mural cell *Pparg* deficiency (Fig. 6d). However, the beneficial effects of Rosi on visceral adipose tissue morphology were largely lost in Mural-*Pparg*^KO animals. Specifically, Rosi treatment did not lead to the accumulation of smaller adipocytes and fewer macrophages in Mural-*Pparg*^KO animals (Fig. 6f, h, j, l, m–p). These data suggest that Rosi triggers healthy visceral WAT remodeling in obese mice by triggering adipocyte differentiation from Pdgfrβ⁺ precursors. Moreover, the tissue mRNA levels of many well-established TZD-regulated genes[61] are not as effectively induced in visceral WAT from Mural-*Pparg*^KO mice (Fig. 6q–r). Importantly, the beneficial effects of Rosi on subcutaneous iWAT morphology (Supplementary Fig. 10d–g) and gene expression (Supplementary Fig. 10h) were not impacted by the loss of *Pparg* in *Pdgfrb*-expressing mural cells; TZDs are likely targeting Pparγ at the level of the adipocyte and/or other cell types in this depot.

Metabolic phenotyping revealed that *Pparg* expression in *Pdgfrb*-expressing cells is essential for the full systemic effects of Rosi on nutrient homeostasis. Rosi treatment did not improve glucose JO!!! Vehicletolerance in obese Mural-*Pparg*^KO mice to the same degree as observed in obese control animals (Supplementary Fig. 11a). Loss of *Pparg* in mural cells also blunted the

ability of Rosi to lower fasting glucose, insulin, and triglyceride levels (Supplementary Fig. 11b–e). It is important to note that the loss of *Pparg* in mural cells only partially blocked the effects of Rosi. This may be explained by the fact that subcutaneous adipocytes, T cells, and other cell types also mediate the beneficial effects of TZDs. In fact, among the various WAT depots, iWAT appears to be the most responsive to the actions of TZDs[61]. Nevertheless, these data suggest that TZDs work in mice, in part, by promoting a healthy remodeling of visceral WAT.

## Discussion

The strong correlation between adipose tissue cellularity and systemic metabolic health in obesity has led to the hypothesis that increasing the number of adipocytes, at the expense of adipocyte hypertrophy, could prevent pathological remodeling and adipose dysfunction[18,22,25,27–29]. In this regard, the recruitment of new adipocytes in response to caloric excess is viewed as a protective mechanism to ensure safe energy storage in WAT and prevent lipotoxicity in peripheral tissues. Recent GWAS analyses lend credence to the notion that impaired adipocyte differentiation in adults may play a causal role in the development of insulin resistance in obesity. Genetic variance at loci containing genes implicated in adipogenesis is associated with insulin resistance and/or ectopic fat accumulation in obese individuals[62,63]. Nevertheless, the importance of adult de novo adipocyte differentiation that is triggered specifically in response to caloric excess has been uncertain.

Here, we employed inducible genetic models to manipulate the adipogenic capacity of adipocyte precursors that naturally give rise to visceral white adipocytes in response to high-fat diet feeding. The *Pdgfrb* promoter is actively expressed in these precursors but not in mature adipocytes. Thus, doxycycline-inducible deletion of *Pparg* in these cells renders these cells incapable of undergoing adipogenesis. Inducible overexpression of *Pparg* is restricted to precursors expressing *Pdgfrb* and thus renders them more competent to undergo adipogenesis. Importantly, as cells differentiate the promoter is subsequently shut off; therefore, mature adipocytes are not genetically altered in this system. As a result, these systems employed here allow us to alter the adipogenic capacity of precursor cells without directly impacting mature adipocytes themselves. Moreover, the ability to temporally control these genetic perturbations allows us to define the importance of these precursors specifically in the context of adult animals. As such, these models differ from the previously reported mouse models of hyperplastic adipose tissue.

It is not possible to rule out other direct functions of *Pparg* in mural cells; however, a number of observations strongly suggest

---

**Fig. 4** Healthy visceral WAT expansion in obesity depends on mural *Pparg* expression. **a** *Pdgfrb*^rtTA; *TRE-Cre; Pparg*^loxP/loxP (Mural-*Pparg*^KO) mice are generated by breeding the *Pdgfrb*^rtTA transgenic mice to animals expressing Cre recombinase under the control of the tet-reponse element (*TRE-Cre*) and carrying floxed *Pparg* alleles (*Pparg*^loxP/loxP). Littermates carrying only *Pdgfrb*^rtTA and *Pparg*^−loxP/loxP alleles (i.e. Cre⁻) were used as the control animals. **b** Control and Mural-*Pparg*^KO mice were fed a standard chow diet until 8 weeks of age before being switched to dox-containing HFD for another 20 weeks. Body weights were measured weekly. Control-HFD, n = 14; Mural-*Pparg*^KO-HFD, n = 13. Data points represent mean + s.e.m. **c** Fat mass and lean mass (normalized to body weight) of control and Mural-*Pparg*^KO mice after 20 weeks of dox-diet feeding. n = 7 per genotype. Bars represent mean + s.e.m. **d** Fat pad weight (normalized to body weight) of control and Mural-*Pparg*^KO mice after 20 weeks of dox-HFD feeding. * denotes p < 0.05 from Student's t-test. Control-HFD, n = 14; Mural-*Pparg*^KO-HFD, n = 13. Bars represent mean + s.e.m. **e–h** Representative immunofluorescence staining of Perilipin (green) and Mac-2 (red) in **e, f** gonadal and **g, h** perirenal WAT paraffin sections obtained from control and Mural-*Pparg*^KO mice after 20 weeks of dox-diet feeding. Scale bar, 200 μm. **i–l** Representative trichrome staining of **i, j** gonadal and **k, l** perirenal WAT paraffin sections obtained from control and Mural-*Pparg*^KO mice after 20 weeks of dox-diet feeding. Scale bar, 200 μm. **m** Number of crown-like structures (Mac-2 positive) in the indicated depots obtained from control and Mural-*Pparg*^KO mice after 20 weeks of dox-HFD feeding. * denotes p < 0.05 from Welch's t-test. n = 24 randomly chosen ×10 magnification fields from six individual animals. Bars represent mean + s.e.m. **n** Average adipocyte size in indicated fat depots from control and Mural-*Pparg*^KO mice after 20 weeks of dox-HFD feeding. * denotes p < 0.05 from Student's t-test. Control-HFD, n = 8; Mural-*Pparg*^KO -HFD, n = 7. Bars represent mean + s.e.m. **o, p** Relative mRNA levels of indicted genes in **o** gonadal and **p** perirenal WAT obtained from control and Mural-*Pparg*^KO mice after 20 weeks of dox-chow diet feeding. * denotes p < 0.05 from Welch's t-test. n = 6 per genotype. Bars represent mean + s.e.m.

that the WAT phenotypes observed here are driven largely by the loss or gain in adipogenic capacity of these precursors, rather than gross defects in vascular integrity. Notably, the local adipose and mural cell phenotypes are restricted to those depots in which *Pdgfrb*-expressing cells naturally undergo adipocyte differentiation (gWAT and pWAT), and only appear in settings when adipocyte differentiation from these cells is naturally triggered (HFD feeding or TZD treatment). The adipose depots in the knockout and transgenic models maintained on a chow diet are

nearly indistinguishable from corresponding depots in control animals; *Pparg* deficiency in *Pdgfrb*-expressing cells does not lead to mural cell death or gross defects in the WAT vasculature. These phenotypes differ significantly from models in which Pdgfrβ[+] adipose precursors cells are ablated entirely[64]. It is also not possible to exclude other *Pparg*-dependent functions of visceral adipose tissue Pdgfrβ[+] cells in the regulation of WAT remodeling. In the case of the Mural-*Pparg*[KO] model, it is notable that the expression levels of multiple collagens are elevated in

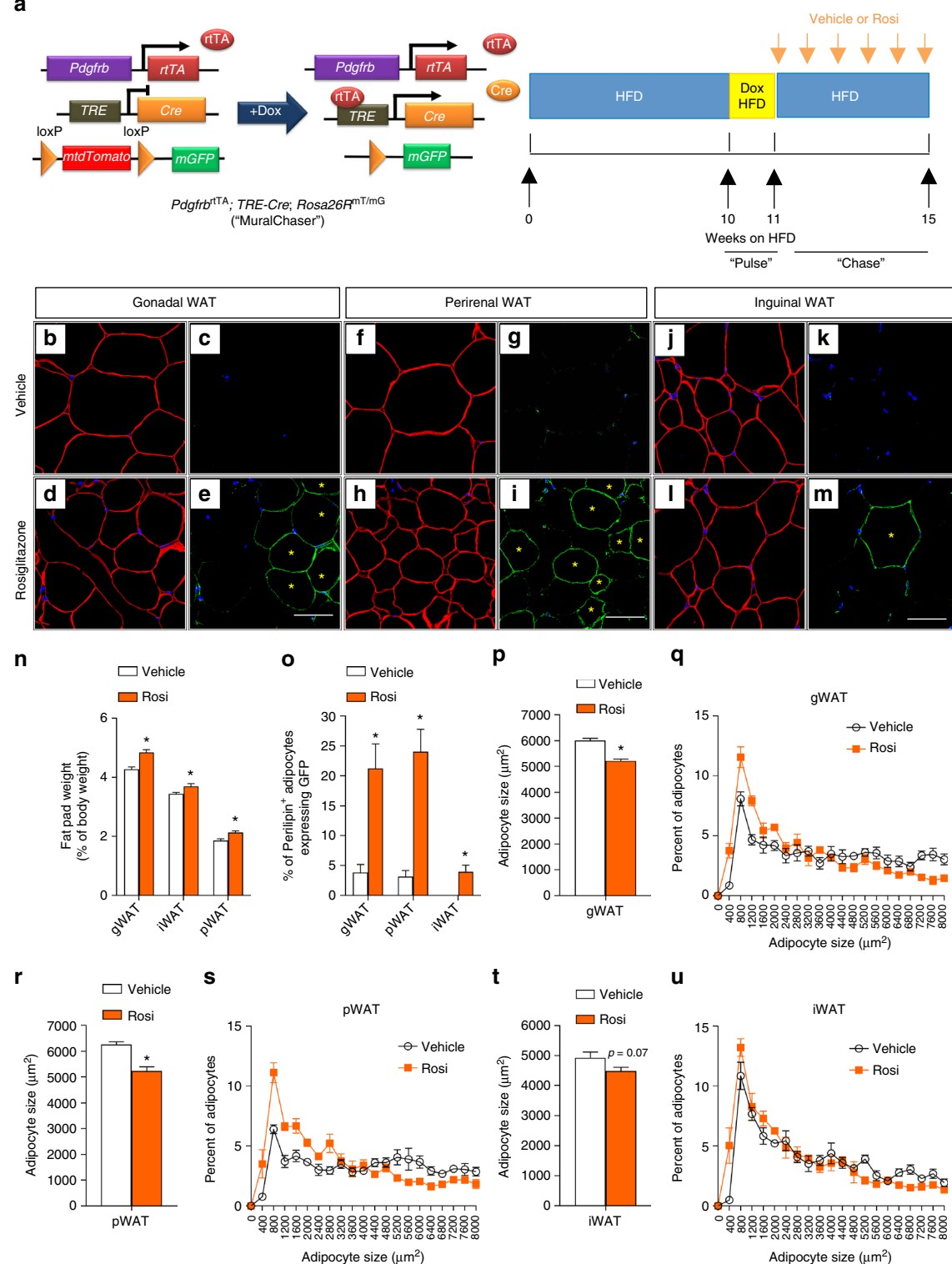

isolated *Pparg*-deficient mural cells from visceral depots. Whether this myofibroblast-like phenotype of Pdgfrβ⁺ cells directly and substantially contributes to local collagen deposition in adipose tissue remains unclear and warrants further investigation. As such, in the case of the Mural-*Pparg*[KO] model, it remains uncertain as to whether the key pathologic feature of the unhealthy visceral WAT is the loss of adipocytes, or the gain in fibrosis and tissue inflammation that ensues when adipogenesis is blocked.

The systemic metabolic phenotypes observed in the models described here are consistent with the notion that the health and functionality of WAT determines systemic metabolic health. The link between healthy WAT and metabolic health has largely been made through studies of subcutaneous adipose tissue. The data presented here extend this link to visceral WAT, in line with recent work from Czech and colleagues[34]. Despite the functional differences between subcutaneous and visceral adipocytes, our data suggest that the health of visceral adipocytes, rather than their abundance or location per se, is a critical determinant of metabolic health in obesity. In fact, increasing the overall number of healthy visceral adipocytes in the setting of caloric excess is associated with improvements in insulin sensitivity in our transgenic model. It is important to note that the presence of *Pdgfrb*-expressing perivascular cells in other tissues that are targeted in these models does limit our ability to unequivocally establish direct causal relationships between visceral WAT health and systemic insulin sensitivity. The observation that the improved insulin sensitivity observed in the Mural-*Pparg*[TG] mice correlates with increased serum adiponectin levels, and depends on the presence of this adipokine, does strongly implicate improved adipocyte function as a major driver of the metabolic phenotype in this model; however, it is still unclear whether increased adiponectin secretion per se directly from the healthy visceral WAT depots of these animals is the primary driver of improved systemic glucose homeostasis. Additional studies are needed to understand the precise mechanisms by which healthy visceral WAT can mediate improvements in glucose homeostasis. In fact, the Mural-*Pparg*[TG] mice described here may be a useful tool to identify adipokines and/or secreted metabolites linked to healthy vs. unhealthy adipocyte function in obesity. Furthermore, one important question that remains is whether adipocytes emerging in response to HFD feeding in adults are molecularly and functionally distinct from pre-existing visceral adipocytes.

A number of adipocyte precursor populations have been recently described[65]; however, it has remained unclear as to which populations are critical for healthy adipose tissue expansion in setting of obesity. Our findings suggest a model in which pathological visceral adipose expansion occurs in DIO mice when the adipogenic capacity of *Pdgfrb*-expressing cells is compromised

(Supplementary Fig. 12). As such, these data provide evidence to the hypothesis that de novo adipocyte differentiation in the setting of caloric excess, even in the visceral depots, is actually a protective mechanism to ensure sustained adipose tissue function[29]. We of course cannot rule out a contribution of other described precursor populations (e.g. non-perivascular Pref-1 or Pdgfrα⁺ cells[66,67]) in regulating WAT growth and expansion. Nevertheless, our studies indicate that the adipogenic capacity of adipose *Pdgfrb*-expressing mural cells is essential for visceral adipose tissue health in mouse obesity. To date, it still remains unclear as to what exact signals drive adipogenesis in the context of diet-induced obesity, and why this occurs in a depot-selective manner. Work from Jeffrey et al.[44] previously highlighted the contribution of sex hormones and the local microenvironment to the regulation of adipogenesis in obesity. This may ultimately play a role in the controlling the differential body fat distribution often observed in males vs. females. Our work here focused exclusively on male animals; additional studies of the female models utilized here will be needed to determine if the contribution of Pdgfrβ⁺ precursors is sex-dependent. Moreover, additional studies will be needed to identify the exact signals and/ or macronutrients associated with high-fat diet feeding that trigger adipogenesis from these precursors.

Our data do indicate that TZDs promote health visceral WAT remodeling in obese mice through stimulation of Pparγ in mural cells. Additional studies of TZD-treated patients will be needed to determine whether short-term treatment with these drugs promotes healthy visceral adipose remodeling, prior to when the shift in body fat distribution occurs. It is notable that *Pdgfrb*-expressing precursors differentiate into adipocytes upon HFD feeding or in response to TZD treatment, but do not contribute significantly to the homeostatic replenishment of adipocytes in lean chow-fed mice. This lends further credence to the emerging concept of adipose precursor heterogeneity.

The data presented here highlight the protective role of de novo adipocyte differentiation in the setting of caloric excess. Moreover, these data provide proof of concept that expanding the number of visceral adipocytes can occur without increasing overall adiposity/body weight per se. Humans maintain functional adipose progenitors and the capacity to form new adipocytes into adulthood[68]. As such, adult adipocyte progenitors may represent a viable target for therapeutic intervention in diabetes.

## Methods

**Animals**. *Pparg*[loxP/loxP] (B6.129-*Pparg*[tm2Rev]/J; JAX 004584), *Rosa26R*[mT/mG] (B6.129(Cg)-*Gt(ROSA)26Sor*[tm4(ACTB-tdTomato,-EGFP)Luo]/J; JAX 007676), and *TRE-Cre* (B6.Cg-Tg(tetO-cre)1Jaw/J; JAX 006234) strains were from Jackson laboratory. *Pdgfrb*[rtTA] transgenic mice (C57BL/6-Tg(Pdgfrb-rtTA)58Gpta/J; JAX 028570) have been described previously[37]. *TRE-Pparg2* transgenic mice were derived by the

---

**Fig. 5** Pdgfrβ⁺ precursors contribute to rosiglitazone-induced adipocyte hyperplasia. **a** MuralChaser mice were fed on standard chow diet until 6 weeks old before being switched to HFD for 10 weeks. Mice were then administrated with dox-containing HFD (600 mg/kg) for 7 days ("pulse"). Following the pulse-labeling period, mice were switched back to regular HFD for another 4 weeks during which vehicle (1% methylcellulose) or rosiglitazone (Rosi) (10 mg/kg/day) was delivered by gavage ("chase"). Paraffin sections of WAT from these animals were stained with anti-GFP (green) and anti-perilipin (red), then counterstained with DAPI (blue; nuclei). **b–e** Representative ×63 magnification confocal immunofluorescence images of gonadal WAT sections after rosiglitazone treatment ("chase"). * indicates GFP-labeled perilipin-positive cells. Scale bar, 50 μm. **f–i** Representative ×63 magnification confocal immunofluorescence images of perirenal WAT sections after rosiglitazone treatment ("chase"). * indicates GFP-labeled perilipin-positive cells. Scale bar, 50 μm. **j–m** Representative ×63 magnification confocal immunofluorescence images of inguinal WAT sections after rosiglitazone treatment ("chase"). * indicates GFP-labeled perilipin-positive cells. Scale bar, 50 μm. **n** Fat pad weight (normalized to body weight) of the indicated fat depots of mice treated with vehicle or rosiglitazone. Student's *t*-test, *$p < 0.05$; $n = 6$ per group. Bars/data points represent mean + s.e.m. **o** Percentage of perilipin-positive adipocytes expressing GFP in the indicated fat depots of mice treated with vehicle or rosiglitazone. Welch's *t*-test, *$p < 0.05$; $n = 12$ randomly chosen ×63 magnification fields from three individual reporter animals. Bars/data points represent mean + s.e.m. **p** Average adipocyte size and **q** distribution of adipocyte size in gonadal WAT from mice treated with vehicle or rosiglitazone. Student's *t*-test, *$p < 0.05$; $n = 6$ per group. Bars/data points represent mean + s.e.m. **r** Average adipocyte size and **s** distribution of adipocyte size in perirenal WAT from mice treated with vehicle or rosiglitazone. Student's *t*-test, *$p < 0.05$; $n = 6$ per group. Bars/data points represent mean + s.e.m. **t** Average adipocyte size and **u** distribution of adipocyte size in inguinal WAT from mice treated with vehicle or rosiglitazone. $n = 6$ per group. Bars/data points represent mean + s.e.m.

UTSW transgenic core facility. All animals used in this study were male and on a pure C57BL/6 background. Mice were maintained with a 12-h light/dark cycle and free access to food and water. All animal studies were performed with permission from the UTSW Institutional Animal Care and Use Committee, and all experiments were conducted according to approved procedures.

**Derivation of *TRE-Pparg2* mice.** *TRE-Pparg2* mice were generated by subcloning of full-length murine *Pparg2* cDNA with an N-terminal FLAG tag into the pTRE

vector (Clontech) (*Hin*dIII and *Sma*I restriction sites). A Kozak sequence consisting of 5′-GCCGCCACC-3′ was inserted in front of the start codon and the rabbit β-globin 3′ untranslated region (UTR) was cloned into the 3′ end of the vector. After linearization (*Sal*I digestion) and purification, *TRE-Pparg2* plasmid was injected into embryos of a pure C57BL/6 background by the Transgenic Core Facility at UT Southwestern Medical Center. Founders and offspring were identified by PCR using the following primers (5′ to 3′):

TrePparg2 F: TCAGGCAGATCGTCACAGAG
TrePparg2 R: tttgccccctccatataaca

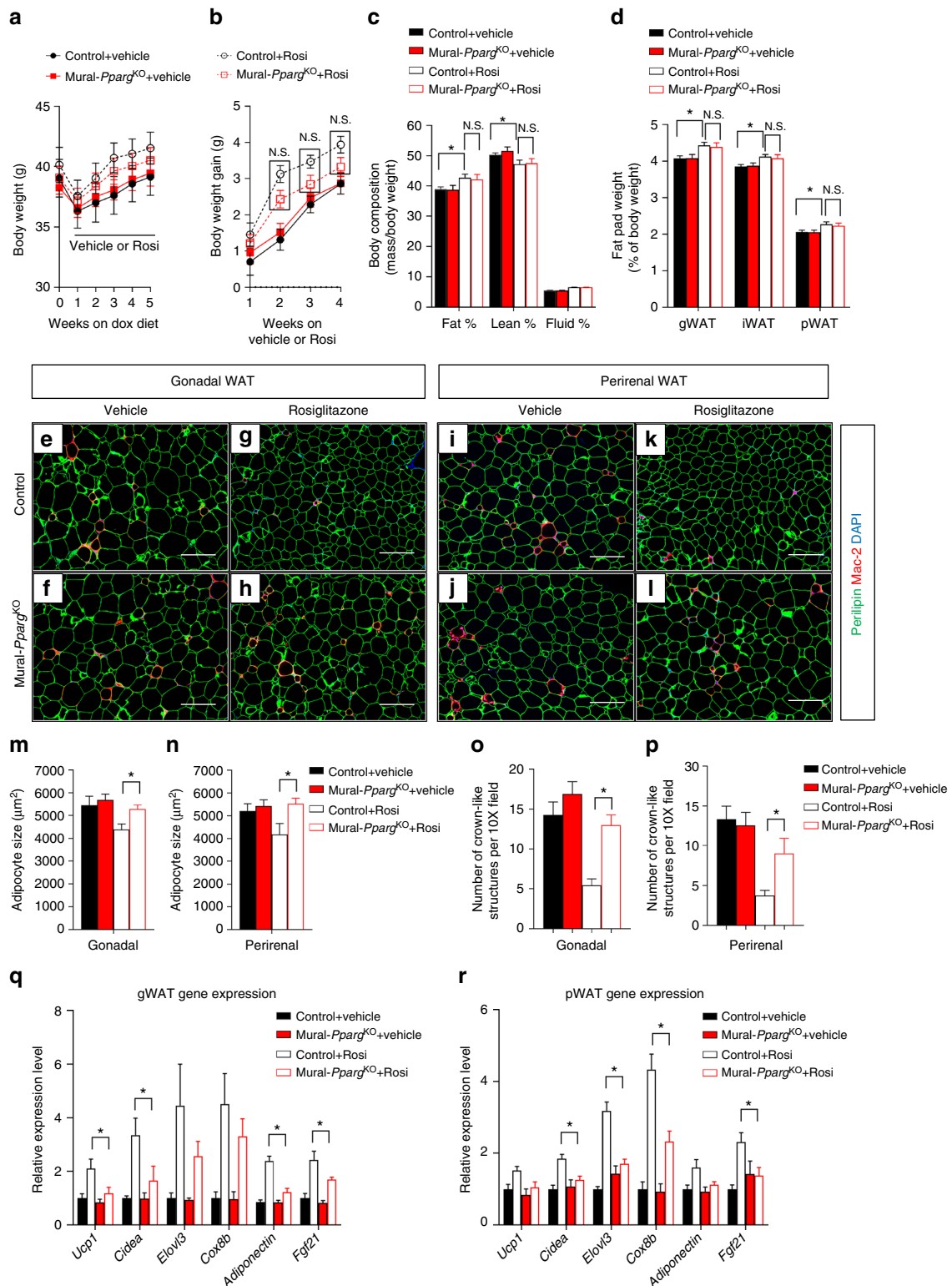

**Rodent diets and drug treatments**. Mice were maintained on a standard rodent chow diet or chow diet containing 600 mg/kg doxycycline (DOX) (Bio-Serv, S4107). For high-fat diet studies, mice were fed a standard high-fat diet (60 kcal% fat; Research Diets, D12492i) or doxycycline-containing high-fat diet (600 mg/kg dox, 60% kcal% fat, Bio-Serv, S5867) as described in the text. For rosiglitazone administration, mice were gavaged daily with vehicle (1% methylcellulose; Fisher Scientific) or 10 mg/kg of rosiglitazone (Cayman Chemical).

**Body composition analysis**. Body fat mass and lean mass were measured in conscious mice using the Bruker Minispec mq10 NMR (UTSW Metabolic Phenotyping Core).

**Serum and liver measurements**. Serum levels of triglycerides, insulin, and adiponectin were determined using a triglyceride determination kit (Sigma, triglyceride reagent T2449 and free glycerol reagent F6428), mouse insulin ELISA kit (Crystal Chem, 90080), and the mouse adiponectin ELISA (Millipore, EZMADP-60K), respectively. For liver triglyceride measurements, ~50 mg of liver tissue was homogenized in phosphate-buffered saline (PBS) and mixed sufficiently with 1.6 ml of CHCL3-CH3OH (2:1, v/v). After centrifugation at 3,000 r.p.m. for 10 min at room temperature, the lower organic phase was transferred and air dried completely in a chemical hood. Samples were re-suspended using 1% Triton X-100 in absolute ethanol and triglycerides were measured using serum triglyceride determination kit (Sigma, triglyceride reagent T2449 and free glycerol reagent F6428).

**Isolation of adipose stromal vascular fraction (SVF) and in vitro differentiation**. White fat depots dissected from mice (4–5 weeks of age) were washed in PBS, minced thoroughly, and digested in vacuum filtered digestion buffer (100 mM Hepes pH 7.4, 120 mM NaCl, 50 mM KCl, 5 mM glucose, 1 mM CaCl2, 1.5% bovine serum albumin, and 1 mg/ml collagenase D) for 2 h at 37 °C with shaking. Digested tissues were then filtered through 100 μM cell strainers to remove undigested tissues. The flow-through was centrifuged for 5 min at 600 g to pellet the stromal vascular cells. The floating adipocyte layer was discarded and the stromal vascular cells were re-suspended in growth media containing DMEM/F12 (Invitrogen) plus 10% fetal bovine serum (FBS). For in vitro differentiation, SVF cells isolated by collagenase digestion were plated onto collagen-coated dishes and cultured in 10% $CO_2$ at 37 °C until confluency. Confluent cultures were stimulated with adipogenic cocktail (growth media supplemented with 5 μg/ml insulin, 1 μM dexamethasone, and 0.5 mM isobutylmethylxanthine) for 48 h. Subsequently, cells were maintained in growth media supplemented with 5 μg/ml insulin until harvest.

**Oil red O staining**. Differentiated cells were fixed in 10% formalin for 10 min at room temperature. Following fixation, the cells were washed with deionized water twice and incubated in 60% isopropanol for 5 min. Cells were completely air dried at room temperature before Oil red O working solution (2 g/l Oil red O in 60% isopropanol) was applied. After incubation at room temperature for 10 min, the Oil red O solution was removed and the cells were washed with deionized water for four times before the images were acquired for analysis.

**Flow cytometry analysis**. SVF cells from the indicated WAT depots were first incubated on ice for 20 min in 2% FBS/PBS containing anti-mouse CD16/CD32 Fc Block (clone 2.4G2) (1:200). Cells were then incubated with primary antibody (anti-CD31 clone 390 1:200, anti-CD45 clone 30-F11 1:200, anti-CD140b clone APB5 1:200) at 4 °C for 30 min. Following three washes of pelleted cells with 2% FBS/PBS, samples were analyzed using a FACS Calibur™ flow cytometer or sorted by a FACS Aria™ flow cytometry at UT Southwestern Medical Center Flow Cytometry Core Facility. All antibodies were obtained commercially from BioLegend (San Diego, CA USA).

**Gene expression analysis**. Total RNA from tissue or cells was extracted and purified using the TRIzol reagent (Invitrogen) and the RNeasy Mini Kit (Qiagen). Total RNA from FACS-sorted cells was extracted using RNAqueous microRNA isolation kit (Thermo Fisher Scientific). cDNA was synthesized with M-MLV reverse transcriptase (Invitrogen) and random hexamer primers (Invitrogen). Relative expression of mRNAs was determined by quantitative PCR using SYBR Green PCR system (Applied Biosystems) and values were normalized to Rps18 levels using the ΔΔ-Ct method. Primers sequences used for quantitative PCR are listed in Supplementary Table 1.

**Histological analysis**. Tissues were dissected and fixed in 4% paraformaldehyde overnight. Paraffin embedding, sectioning, and trichrome staining were performed at the Molecular Pathology Core Facility at UTSW. Bright-field and fluorescent images were acquired using a Keyence BZ-X710 microscope. Adipocyte size analysis was conducted as described[69]. Keyence BZ-X Analyzer software was used for analysis of bright-field images of H&E-stained paraffin sections. >200 adipocytes were quantified in each individual animal.

**Indirect immunofluorescence**. The following antibodies and concentrations were used: guinea pig anti-perilipin 1:1500 (Fitzgerald 20R-PP004); rabbit anti-Mac2 1:500 (Cedarlane, Clone M3/38); chicken anti-GFP 1:700 (Abcam, ab13970); donkey anti-rabbit Alexa 647 1:200 (Invitrogen, A-31632); goat anti-guinea pig Alexa 488 1:200 (Thermo Fisher Scientific, A-11073); goat anti-chicken Alexa 488 1:200 (Invitrogen, A-11039); and goat anti-guinea pig Alexa 647 1:200 (Invitrogen, A-21450). Paraffin sections were dewaxed and hydrated in xylene and 100–95–80–70–50% ethanol and $ddH_2O$. Slides were placed in chambers containing 1× R-Buffer A pH 6.0 solution and antigen retrieval was performed using Antigen Retriever 2100 (Electron Microscopy Sciences) for 2 h. Following one PBS wash for 5 min, Fx Signal Enhancer (Invitrogen) was added to the slides for 30 min at room temperature. Slides were then blocked for 30 min in PBS containing 10% normal goat serum at room temperature. Primary antibodies were diluted in PBS containing 10% normal goat serum and added to paraffin sections overnight at 4 °C. Following overnight incubation, slides were washed in PBS and incubated with secondary antibodies diluted in PBS containing 10% normal goat serum for 2 h at room temperature. Washed slides were mounted with Prolong Anti-Fade mounting medium containing DAPI (Invitrogen) before images were acquired for analysis.

**In vivo insulin stimulation**. After 6-h fasting, animals were anesthetized and injected with insulin (2 U/kg) through the portal vein. Five minutes after the injection, gWAT, iWAT, liver, and soleus muscle were quickly snap-frozen for subsequent analysis. Samples harvested before insulin injection were used as untreated controls.

**Immunoblotting and antibodies**. Protein extracts from cells or tissues were prepared by homogenization in RIPA lysis buffer (Santa Cruz). Protein extracts were separated by SDS-polyacrylamide gel electrophoresis (SDS-PAGE) and transferred onto PVDF membrane (Millipore). After incubation with the indicated primary antibodies at 4 °C overnight, the blots were incubated with IR dye-coupled secondary antibodies (LI-COR) and visualized by the LI-COR Odyssey infrared imaging system. The anti-FLAG (F3165) and anti-β-actin (A1978) antibodies were from Sigma. The anti-phospho-Akt (Ser473) (9271) and anti-Akt (2920) antibodies were from Cell Signaling Technology. Full-length western blot images corresponding to the data presented in main/supplementary figures are provided in Supplementary Figure 13.

**Fig. 6** Mural cell *Pparg* is required for TZD-driven visceral WAT remodeling. **a**, **b** Control and Mural-*Pparg*[KO] mice were treated as described in Supplemental Figure 10a. Body weights were measured weekly. Control + Vehicle, n = 8; Mural-*Pparg*[KO] + Vehicle, n = 9; Control + Rosi, n = 11; Mural-*Pparg*[KO] + Rosi, n = 19. n.s. denotes not statistically significant from two-way ANOVA. Data points represent mean + s.e.m. **c** Fat mass and lean mass (normalized to body weight) of control and Mural-*Pparg*[KO] mice after 4 weeks of vehicle or rosiglitazone treatment. Control + Vehicle, n = 8; Mural-*Pparg*[KO] + Vehicle, n = 8; Control + Rosi, n = 8; Mural-*Pparg*[KO] + Rosi, n = 9. * denotes p < 0.05 from Student's t-test. n.s. denotes not statistically significant. Bars represent mean + s.e.m. **d** Fat pad weight (normalized to body weight) of control and Mural-*Pparg*[KO] mice after 4 weeks of vehicle or rosiglitazone treatment. Control + Vehicle, n = 8; Mural-*Pparg*[KO] + Vehicle, n = 8; Control + Rosi, n = 8; Mural-*Pparg*[KO] + Rosi, n = 9. * denotes p < 0.05 from Student's t-test. n.s. denotes not statistically significant. Bars represent mean + s.e.m. **e–l** Representative immunofluorescence staining of Perilipin (green) and Mac-2 (red) in **e–h** gonadal and **i–l** perirenal WAT paraffin sections obtained from control and Mural-*Pparg*[KO] mice after 4 weeks of vehicle or rosiglitazone treatment. Scale bar, 200 μm. **m**, **n** Average adipocyte size in **m** gonadal and **n** perirenal WAT from control and Mural-*Pparg*[KO] mice treated with vehicle or rosiglitazone. Two-way ANOVA, *p < 0.05; Control + Vehicle, n = 6; Mural-*Pparg*[KO] + Vehicle, n = 6; Control + Rosi, n = 7; Mural-*Pparg*[KO] + Rosi, n = 8. Bars represent mean + s.e.m. **o**, **p** Number of crown-like structures (Mac-2 positive) in **o** gonadal and **p** perirenal WAT from control and Mural-*Pparg*[KO] mice treated with vehicle or rosiglitazone. Two-way ANOVA, *p < 0.05. n = 24 randomly chosen ×10 magnification fields from six individual animals. Bars represent mean + s.e.m. **q**, **r** Relative mRNA levels of indicated rosiglitazone-induced genes in **q** gonadal and **r** perirenal WAT obtained from control and Mural-*Pparg*[KO] mice treated with vehicle or rosiglitazone. Two-way ANOVA, *p < 0.05. n = 6 per group. Bars represent mean + s.e.m.

**Metabolic phenotyping.** For glucose tolerance tests (GTT), mice were injected i.p. with glucose (Sigma) at the dosage of 1 g per kg body weight after an overnight fast. For insulin tolerance test (ITT), mice were injected i.p. with 0.75 U human insulin (Eli Lilly) per kg body weight after a 6 h fast. Blood was collected by venous bleeding from the tail vein at 0, 15, 30, 60, 90, and 120 min post injection. Glucose concentrations were measured using Bayer Contour glucometers. Hyperinsulinemic–euglycemic clamps were performed on conscious, unrestrained mice as previously described[70].

**Statistics.** No statistical method was used to predict samples sizes. Sample sizes were determined on the basis of the experience and reported experimental designs. No randomization or blinding was performed to allocate the samples. No criteria of inclusion or exclusion of data were used. Statistical analysis was carried out as indicated in figure legends. Data variance was examined by $F$-test or Bartlett's test. Data normality was test by Shapiro–Wilk test. The data meet the assumptions of the indicated statistical analysis. All tests were performed as two sided. Results with a $p$-value less than 0.05 were considered significant. All quantitative data were shown as mean ± s.e.m. All statistics were calculated using GraphPad Prism7 or Microsoft Excel. All statistical data are provided in Supplementary Data 1.

**Data availability.** The data that support the findings of this study are available from the corresponding author upon reasonable request.

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

## Acknowledgements

The authors are grateful to the members of the UTSW Touchstone Diabetes Center for useful discussions, and Drs. P. Scherer and C. Kusminski for critical reading of the manuscript. The authors thank the UTSW Animal Resource Center, Metabolic Pheno-typing Core, Transgenic Core, Pathology Core, Live Cell Imaging Core, and Flow Cytometry Core for excellent guidance and assistance with experiments performed here. This study was supported by the Searle Scholars Program (Chicago, IL), American Heart Association (AHA) 15BGIA22460021, American Diabetes Association 1-17-IBS-181, and NIDDK R01 DK104789 to R.K.G., NIDDK R00-DK094973 and JDRF Award 5-CDA-2014-185-A-N to W.L.H., AHA postdoctoral fellowship 16POST26420136 to M.S., NIH NIGMS training grant T32 GM008203 and NIDDK F31 DK113696-01 to C.H., and NIDDK K99 DK114498 to Y.Z.

## Author contributions

M.S. and R.K.G. conceived, designed, and analyzed the experiments and wrote the manuscript. M.S., L.V., N.C.B., C.H., B.S., Y.A.A., and Y.Z. performed the experiments and analyzed the data. A.X.S., S.C., X.Y., and W.L.H performed the CLAMP experiment and analyzed the data.

## Additional information

**Competing interests:** The authors declare no competing financial interests.

