## [Peer Review File · Nature Communications]

Editorial Note: This manuscript has been previously reviewed at another journal that is not operating a transparent peer review scheme. This document only contains reviewer comments and rebuttal letters for versions considered at Nature Communications. Mentions of prior referee reports have been redacted.

Reviewers' Comments:

Reviewer #1:

Remarks to the Author:

The authors have responded to my concerns in a scholarly and comprehensive manner.

Specifically, the results of new experiments that address metabolism in the absence of adiponectin and tissue specific insulin sensitivity by clamp and Akt phosphorylation provide compelling support for the premise.

In its current form, this manuscript represents a convincing and conceptual advance.

Reviewer #2:

Remarks to the Author:

This work by Gupta and colleagues was previously reviewed by Nature. In the revised manuscript, the authors have very carefully addressed all major points raised by the reviewers, including revisions of the conclusions that can be drawn from the experiments. The manuscript is very interesting and well written. Below are a few points that should be addressed.

The authors have added an additional animal model, mural-PpargTG bred to adiponectin-deficient mice and show that the improvement of glucose tolerance and insulin sensitivity are dependent on adiponectin. They use this as evidence that the improved metabolic function in mural-PpargTG is dependent on adipose tissue function. However, this is a rather indirect piece of evidence, e.g. it is unclear whether adiponectin is actively involved in the "signal" elicited by the increased level of mural PPAR γ or whether it is just required for the insulin sensitizing function of increased mural PPAR γ . I don't think the authors need to outline the mechanism, but they should modify their claims.

It is also not clear which adipose depot secretes increased adiponectin levels in response to ectopic mural-PPAR γ , and it cannot be concluded that it is from the visceral depots. Did the authors investigate whether Pdgfr β + cells in other tissues start to express adipocyte marker genes? Does mural PPAR γ affect the bone marrow?

The authors assume (e.g. page 8) that they can express PPAR γ and "increase the adipogenic capacity of mural cells without directly manipulating adipocyte gene expression per se." That sounds impossible! Ectopic PPAR γ will obviously have major effects on gene expression.

Reviewer #3:

Remarks to the Author:

Authors addressed most of my concerns, except point 4.

Original remark:

4. In Supplementary fig 6c, authors find no changes in Pparg in mature adipocytes after Dox treatment, concluding that the gene expression of the adipocytes is not altered, thus enabling them to study the importance of the adipocytes formation without changes in gene expression. An alternative explanation would be that 7 days is simply not long enough after the Dox treatment for the Pparg overexpressing cells to differentiate in vivo – especially in

the context of the lengths of the studies where authors used HFD for several weeks. How are the total Pparg levels at the end of the experiments, and how is the Pparg expression in the GFP+ cells (from the lineage tracing experiment, eg. shown in Fig 2s,t)?

Authors response:

We thank the reviewer for this question. This is an important point worth emphasis. Pdgfrb is not expressed in mature adipocytes- this is apparent from the new qPCR data in Supplementary Figure 2D. Activation of the TRE-Pparg2 transgene is dependent on the presence of rtTA, which in turn, is dependent on active Pdgfrb expression. As such, as cells differentiate into adipocytes, the Pdgfrb is shut off and rtTA is no longer expressed. On the other hand, mGFP expression in this system is coming from the Rosa26R locus; once Cre mediated activation of the Rosa locus occurs, mGFP expression is constitutive and no longer dependent on Pdgfrb expression. Of course, new adipocytes (mGFP+) emerging after HFD feeding may differ in gene expression from pre-existing adipocytes or adipocytes from control animals; however, this would not be attributed to active transgene expression.

Additional remark:

This explanation describes how their transgenic model works, and this was clear in the original manuscript, but it does not answer my concern. The question is not whether new and pre-existing adipocytes might have different gene expression, but whether the new adipocytes that originated due to Pparg2 overexpression in their precursors differ compared to the naturally occurring new adipocytes in the control mice due to HFD.

REVIEWERS' COMMENTS:

Reviewer #1 (Remarks to the Author):

The authors have responded to my concerns in a scholarly and comprehensive manner.

Specifically, the results of new experiments that address metabolism in the absence of adiponectin and tissue specific insulin sensitivity by clamp and Akt phosphorylation provide compelling support for the premise.

In its current form, this manuscript represents a convincing and conceptual advance.

We thank the reviewer for taking the time to review the manuscript and provide very constructive feedback.

Reviewer #2 (Remarks to the Author):

This work by Gupta and colleagues was previously reviewed by Nature. In the revised manuscript, the authors have very carefully addressed all major points raised by the reviewers, including revisions of the conclusions that can be drawn from the experiments. The manuscript is very interesting and well written. Below are a few points that should be addressed.

We thank the reviewer for carefully reviewing the manuscript and for providing very constructive feedback.

The authors have added an additional animal model, mural-PpargTG bred to adiponectin-deficient mice and show that the improvement of glucose tolerance and insulin sensitivity are dependent on adiponectin. They use this as evidence that the improved metabolic function in mural-PpargTG is dependent on adipose tissue function. However, this is a rather indirect piece of evidence, e.g. it is unclear whether adiponectin is actively involved in the “signal” elicited by the increased level of mural PPARγ or whether it is just required for the insulin sensitizing function of increased mural PPARγ. I don’t think the authors need to outline the mechanism, but they should modify their claims.

This is an excellent point. We have now added/modified the following text of the discussion section (Pages 22-23) to explicitly state this caveat:

“The observation that the improved insulin sensitivity observed in the Mural-Pparg^{TG} mice correlates with increased serum adiponectin levels, and depends on the presence

of this adipokine, does strongly implicate improved adipocyte function as a major driver of the metabolic phenotype in this model; however, it is still unclear whether increased adiponectin secretion *per se* directly from the healthy visceral WAT depots of these animals is the primary driver of improved systemic glucose homeostasis. Additional studies are needed to understand the precise mechanisms by which healthy visceral WAT can mediate improvements in glucose homeostasis. In fact, the Mural-*Pparg*^{TG} mice described here may be a useful tool to identify adipokines and/or secreted metabolites linked to healthy vs. unhealthy adipocyte function in obesity. Furthermore, one important question that remains is whether adipocytes emerging in response to HFD feeding in adults are molecularly and functionally distinct from preexisting visceral adipocytes.”

It is also not clear which adipose depot secretes increased adiponectin levels in response to ectopic mural-PPAR γ , and it cannot be concluded that it is from the visceral depots. Did the authors investigate whether Pdgfr β ⁺ cells in other tissues start to express adipocyte marker genes? Does mural PPAR γ affect the bone marrow?

These are also great questions. As it relates to bone marrow specifically, we are currently investigating whether *Pdgfrb* expression identifies adipocyte precursors giving rise to adipocytes residing within bone marrow, and whether TZDs trigger bone marrow adipocyte differentiation through these cells. This is on-going study that will be published in the future.

The authors assume (e.g. page 8) that they can express PPAR γ and “increase the adipogenic capacity of mural cells without directly manipulating adipocyte gene expression per se.” That sounds impossible! Ectopic PPAR γ will obviously have major effects on gene expression.

We apologize for not making this point more clear. Our intention was to highlight that the transgene will be expressed in Pdgfr β ⁺ precursors but not mature adipocytes (which are Pdgfr β negative). We have modified the statement on page 8 to read:

“As a result, this model enables us to assess the consequences of increasing the adipogenic capacity of mural cells, without directly manipulating the expression of genes in mature adipocytes.”

Reviewer #3 (Remarks to the Author):

Authors addressed most of my concerns, except point 4.

Original remark:

4. In Supplementary fig 6c, authors find no changes in Ppar γ in mature adipocytes after Dox treatment, concluding that the gene expression of the adipocytes is not altered, thus enabling them to study the importance of the adipocytes formation without changes in gene expression. An alternative explanation would be that 7 days is simply not long enough after the Dox treatment for the Ppar γ overexpressing cells to differentiate in vivo – especially in the context of the lengths of the studies where authors used HFD for several weeks. How are the total Ppar γ levels at the end of the experiments, and how is the Pparg expression in the GFP $^{+}$ cells (from the lineage tracing experiment, eg. shown in Fig 2s,t)?

Authors response:

We thank the reviewer for this question. This is an important point worth emphasis.

Pdgfrb is not expressed in mature adipocytes- this is apparent from the new qPCR data

in Supplementary Figure 2D. Activation of the TRE-Pparg2 transgene is dependent on

the presence of rtTA, which in turn, is dependent on active Pdgfrb expression. As such,

as cells differentiate into adipocytes, the Pdgfrb is shut off and rtTA is no longer expressed. On the other hand, mGFP expression in this system is coming from the

Rosa26R locus; once Cre mediated activation of the Rosa locus occurs, mGFP expression is constitutive and no longer dependent on Pdgfrb expression.

Of course, new adipocytes (mGFP $^{+}$) emerging after HFD feeding may differ in gene

expression from pre-existing adipocytes or adipocytes from control animals; however,

this would not be attributed to active transgene expression.

Additional remark:

This explanation describes how their transgenic model works, and this was clear in the original manuscript, but it does not answer my concern. The question is

not whether new and pre-existing adipocytes might have different gene expression, but whether the new adipocytes that originated due to Pparg2 overexpression in their precursors differ compared to the naturally occurring new adipocytes in the control mice due to HFD.

We thank the reviewer for clarifying his/her question. This is indeed a good point. We did not perform a comprehensive global gene expression analysis of isolated adipocytes from the transgenic vs. control animals; however, we do indeed suspect that there would be differences in gene expression that reflect the overall improved health of the visceral WAT in transgenic mice. As part of a new/follow-up study, we are performing an RNA-seq of the adipocytes from these animals, with the hope of finding “signals” that may mediate the improved WAT phenotype. Our point of emphasis here is that these alterations to the adipocyte program would be secondary effects as the transgene is no longer operative in mature cells. We have now amended the following portion of the discussion section to raise the reviewer’s question as a future direction of this work:

“The observation that the improved insulin sensitivity observed in the Mural-*Pparg*^{TG} mice correlates with increased serum adiponectin levels, and depends on the presence of this adipokine, does strongly implicate improved adipocyte function as a major driver of the metabolic phenotype in this model; however, it is still unclear whether increased adiponectin secretion *per se* directly from the healthy visceral WAT depots of these animals is the primary driver of improved systemic glucose homeostasis. Additional studies are needed to understand the precise mechanisms by which healthy visceral WAT can mediate improvements in glucose homeostasis. In fact, the Mural-*Pparg*^{TG} mice described here may be a useful tool to identify adipokines and/or secreted metabolites linked to healthy vs. unhealthy adipocyte function in obesity. Furthermore, one important question that remains is whether adipocytes emerging in response to HFD feeding in adults are molecularly and functionally distinct from preexisting visceral adipocytes.”